# SELF-TRAINING ON UNPAIRED DATA IMPROVES MULTI-MODAL ALIGNMENT

## ABSTRACT

In the past few years, multimodal foundation models, *e.g.*, CLIP, learned from a massive amount of paired multimodal data, emerged and exhibited impressive cross-modal ability in many applications. Yet collecting high-quality paired data is generally costly or even infeasible in certain cases, and the amount of paired multimodal data is several orders fewer than that of *unpaired* unimodal data, *i.e.*, data without any correspondence. Our work focuses on alleviating the excessive demand for paired language-image data by leveraging the abundant unpaired data. We introduce a new approach for vision-language alignment, which we call Language-Image Self-Training (LIST). LIST consists of two key ingredients that function in a synergistic loop: i) a captioner model trained alternatively with the augmented paired data and the unpaired data with synthetic captions, both derived from the data engine, and ii) a data engine that synthesizes a diverse spectrum of captions for both paired and unpaired images with the captioner, integrating synthetic captions with the web-scraped ones to enhance the quality of paired data using off-the-shelf Large Language Models. We observe that the LIST methodology not only significantly improves the alignment between vision and language representations across multiple major benchmarks—zero-shot image classification, image-text retrieval, and compositional evaluation—but also demonstrates strong generalization to audio-language representation alignment.

## 1 INTRODUCTION

Over the last decade, we have witnessed remarkable leaps in the realm of multi-modal foundation models (Radford et al., 2021; Yu et al., 2022; Alayrac et al., 2022; Huang et al., 2023; Radford et al., 2023). These models, with their ability to process and integrate inputs from different modalities, including but not limited to image, language, and audio, have unveiled a realm of unprecedented possibilities. Notably, empowered by contrastive language-image pretraining (CLIP) (Radford et al., 2021), diffusion denoising objective (Ho et al., 2020), and Transformers architecture (Vaswani et al., 2017; Dosovitskiy et al., 2020), multi-modal foundation models have demonstrated remarkable capabilities. They can generate visually appealing images from textual descriptions (Ramesh et al., 2021; Rombach et al., 2022), respond to human instructions conditioned on input images (Li et al., 2022; Alayrac et al., 2022), and synthesize audio from unseen text description (Huang et al., 2023), showing astonishing scalability relative to the volume of data and compute.

However, a caveat accompanies these advances. Multi-modal foundation models depend extensively on *paired* multi-modal data, as evidenced by the use of vast quantities of image-text pairs, reaching into the millions (Radford et al., 2021) or billions (Schuhmann et al., 2022), for training models like CLIP. This reliance poses significant challenges in data acquisition at scale. In practice, the availability of paired multi-modal data is substantially overshadowed by the abundance of unpaired unimodal data, making the former more challenging to collect and access. In the context of image-text datasets, while web-scraping can somewhat ease the collection process, the resulting data often suffers from noise and necessitates extensive efforts in data cleaning. Several studies have emerged to refine the image-text dataset quality. Radenovic et al. (2023) proposed a filtering method to remove noisy image-text pairs, Fan et al. (2023) explored rewriting captions using large language models, and Nguyen et al. (2023); Lai et al. (2023); Yu et al. (2024) suggested enhancing raw captions with those synthesized by a specialized captioner. However, these approaches still fundamentally rely on the assumption of having access to a substantial volume of data pairs beforehand.

Figure 1: **Language-Image Self-Training (LIST).** It consists of two key ingredients that work synergistically as a loop: i) a **captioner model** trained alternatively with the augmented paired data and the synthetic paired data derived from the data engine, and ii) a **data engine** that synthesizes diversified captions for both paired and unpaired images with the captioner, while merging synthetic and web-scraped captions to enhance the quality of paired data with off-the-shelf Large Language Models. The dashed arrow indicates that the captioner is not involved in the first step of the loop.

This study aims to mitigate the limitations inherent in relying on paired data, aligning different modalities by leveraging the untapped potential of unpaired data. We present *Language-Image Self-Training (LIST)*, a new methodology for vision-language alignment. As depicted in Fig. 1, LIST comprises two essential elements, the *captioner model* and the *data engine*, that work in a dynamic, synergistic loop:

1. **Captioner model.** This model, instantiated by Transformers (Vaswani et al., 2017; Dosovitskiy et al., 2020), is alternately trained using two types of data. First is a small-scale *paired data*, augmented by the data engine with the help of LLMs, *e.g.*LLaMa (Touvron et al., 2023). The second one is the *unpaired data*, each of which is uniquely paired with multiple synthetic captions synthesized by the data engine. Both data types are sourced from our sophisticated data engine, providing diverse and comprehensive training supervision.

2. **Data engine.** This engine is tasked with generating a wide array of captions for both paired and unpaired images, using the captioner model. In addition, for the paired data, the data engine integrates synthetic captions with those scraped from the web. This integration process is supported by advanced off-the-shelf LLMs, ensuring high-quality and contextually appropriate captions.

Together, these components enable LIST to effectively incorporate unpaired data to train vision-language models, requiring only a small amount of data pairs to warm up the training and allowing for the explicit control of the captions synthesis. In addition, since LIST does not have any design specific to the vision input, it can be easily generalized to broader scenarios where paired data are scarce. For instance, much like the transition from CLIP (Radford et al., 2021) to CLAP (Elizalde et al., 2023), LIST demonstrates strong generalization beyond the vision-language tasks to the audio-language setting, where typically only a few thousand paired examples are available.

In summary, this paper makes several significant contributions to multi-modal alignment:

- We introduce *Language-Image Self-Training (LIST)*, a generic framework designed to harness the untapped potential of unpaired data for enhancing multi-modal representation alignment. LIST exhibits astonishing performance across multiple benchmarks, including zero-shot image/audio classification, image-text retrieval, and compositional evaluation.

- Our approach demonstrates how the integration of a captioner model with a data engine, operating in a synergistic loop within LIST, can lead to concurrent improvements in both the model's performance and the quality of the data.

- The data engine in LIST is able to generate a diverse range of captions for both paired and unpaired images. By leveraging LLMs, it effectively integrates the information of web-scraped and synthetic captions, thereby enhancing the quality of the paired data.

## 2 RELATED WORK

**Multimodal foundation models.** Multimodal foundation models (Radford et al., 2021; Jia et al., 2021; Yu et al., 2022; Alayrac et al., 2022; Li et al., 2022; Liu et al., 2023) have exhibited remarkable capabilities in understanding and generating outputs across various modalities. CLIP (Radford et al., 2021) uniquely processes images and text as parallel data streams, enabling seamless connections between visual and textual content. This approach is further advanced by models like DALL-E (Ramesh et al., 2021) and StableDiffusion (Rombach et al., 2022), which extend the concept to generate intricate images from textual descriptions, and (Guzhov et al., 2022; Huang et al., 2023) that extend it to audio understanding. These models rely heavily on extensive paired datasets for training and have demonstrated notable proficiency in diverse cross-modal tasks, including image captioning, text-to-image synthesis, and audio-visual correlations. This surge in multimodal learning highlights the importance of large-scale, diverse datasets. However, the challenge lies in the labor-intensive process of curating high-quality image-text pairs, especially at scale. This work seeks to mitigate the reliance on paired data by capitalizing on the abundance of readily available unpaired data, utilizing self-training techniques to bridge this gap.

**Improving multi-modal datasets.** Multi-modal datasets, sourced primarily through internet crawling, are susceptible to noise and biases due to limited human moderation, often reflecting a narrow spectrum of human interests. Recent research has focused on improving the quality of vision-language pretraining datasets by caption filtering (Radenovic et al., 2023; Cao et al., 2023), caption rewriting (Fan et al., 2023), using synthetic caption (Li et al., 2022; 2023a; Santurkar et al., 2022), raw-synthetic caption mixing (Nguyen et al., 2023), and image synthesis (Tian et al., 2023) with text-to-image diffusion model (Rombach et al., 2022). Closer to LIST, concurrent methods, *e.g.*, CapsFusion (Yu et al., 2024), VeCLIP (Lai et al., 2023), and ALIP (Yang et al., 2023a), also explored generating synthetic captions and then merging them with the web-scraped ones. However, they are highly dependent on external VLMs trained on additional paired data, while LIST does not have access to any implicit source of data pairs. These enhancements have shown that optimized datasets can significantly boost model performance, making it possible for models trained on fewer, but higher-quality, text-image pairs to match or even surpass those trained on larger datasets. LIST stands apart in its fundamental aim to *leverage unpaired data rather than enhancing paired data*, an approach that ideally complements the existing methodologies.

**Self-training.** Self-training (Scudder, 1965; Fralick, 1967; Blum & Mitchell, 1998), a semi-supervised learning technique, has emerged as a significant approach for its effective use of unlabeled data to boost model performance. This methodology, which entails generating pseudo-labels for unlabeled data using the model itself, has demonstrated considerable improvements in performance. It proves beneficial not only in areas with limited labeled data (Broder, 1997; Lee et al., 2013) but also in larger-scale applications (He et al., 2019; Yalniz et al., 2019; Xie et al., 2020). Recent advancements in self-training have extended its application to various domains, such as language modeling (Huang et al., 2022a; Dong et al., 2023; Gulcehre et al., 2023), LLMs alignment (Li et al., 2023b), and object detection (Zoph et al., 2020), making it invaluable where labeled data acquisition is challenging or costly. LIST aligns with this paradigm but distinguishes itself through its task for vision-language alignment and the unique caption refinement process that incorporates external knowledge from LLMs. In a related context, UCM (Yang et al., 2023b) similarly proposes a self-training method for vision-language BERT, albeit with a different emphasis than vision-language alignment. Nonetheless, UCM's methodology is heavily dependent on an external object detector, while LIST is a self-reliant framework, functioning independently even in the absence of the LLMs–the sole external component LIST utilizes. Furthermore, leveraging LLMs enables LIST to integrate open-set knowledge, which is a significant enhancement over the object detector used in UCM that is constrained by a predefined set of close-set classes.

## 3 LANGUAGE-IMAGE SELF-TRAINING

### 3.1 THE INTERPLAY OF MODEL AND DATA

**Notation.** Let $\boldsymbol{x}, \boldsymbol{y}$ denote image and caption respectively, $\mathcal{D}_p = \{(\boldsymbol{x}_i, \boldsymbol{y}_i)\}_{i=1}^{N_p}$ denote a paired dataset consisting of $N_p$ image-caption pairs, $\mathcal{D}_u = \{\boldsymbol{x}_i\}_{i=1}^{N_u}$ denote a dataset consisting of $N_u$

unpaired images ($N_u \gg N_p$), a vison-language model denote $\mathbf{M}(\cdot)$ that can process both image and text, and a data engine $\mathcal{E}$ that synthesizes captions with the help of $\mathbf{M}$ and (optionally) an LLMs $\mathbf{G}$.

**Framework.** We introduce *Language-Image Self-Training (LIST)*, a generic framework designed to utilize paired data $\mathcal{D}_p$ as well as harness the untapped potential of unpaired data $\mathcal{D}_u$ for enhancing vision-language model $\mathbf{M}$, and vice versa. As shown in Fig. 1, LIST operates in a loop, trains the vision-language model alternatively on $\mathcal{D}_p$ and $\mathcal{D}_u$, augmented by the data engine $\mathcal{E}$. Here we briefly present a high-level description of the loop of LIST, leaving the details to the next two subsections for clarity:

1. We start the cycle by augmenting the initial, small-scale paired dataset $\mathcal{D}_p$ with the data engine $\mathcal{E}$ as:

$$\mathcal{E}(\mathcal{D}_p;\ \mathbf{G}) = \{(\boldsymbol{x}_i,\ \mathcal{E}(\boldsymbol{y}_i;\ \mathbf{G}))\}_{i=1}^{N_p} \tag{1}$$

$$= \{(\boldsymbol{x}_i,\ \{\hat{\boldsymbol{y}}_i^j : \hat{\boldsymbol{y}}_i^j \sim \mathbf{G}(\boldsymbol{y})\}_{j=1}^m)\}_{i=1}^{N_p}, \tag{2}$$

where the data engine $\mathcal{E}$ takes an image-text pairs as input and generate $m$ captions $\{\hat{\boldsymbol{y}}^j\}_m$ that are augmented by the LLM $\mathbf{G}$ for each image. The details of the prompt design will be given in the next subsection.

2. Given the augmented paired dataset $\mathcal{E}(\mathcal{D}_p)$, we train the vision-language model $\mathbf{M}$ with Empirical Risk Minimization:

$$\mathbf{M}_p = \arg\min_{\mathbf{M}}\ \mathbb{E}_{(\boldsymbol{x},\boldsymbol{y})\sim\mathcal{E}(\mathcal{D}_p)}\ \mathcal{L}(\boldsymbol{x},\boldsymbol{y};\mathbf{M}), \tag{3}$$

where $\mathcal{L}(\cdot)$ is an objective function that will be given subsequently and the subscript of $\mathbf{M}$ differentiates the models trained with paired data ($\mathbf{M}_p$) and unpaired data ($\mathbf{M}_u$). Since there are $m$ captions associated with an image in $\mathcal{D}_p$, we sample one of them uniformly at random.

3. Now, with $\mathbf{M}_p$ to empower $\mathcal{E}$, we can then synthesize captions for all the images in $\mathcal{D}_u$ as:

$$\mathcal{E}(\mathcal{D}_u;\mathbf{M}_p) = \{\boldsymbol{x}_i,\ \mathcal{E}(\boldsymbol{x}_i;\ \mathbf{M}_p)\}_{i=1}^{N_u} \tag{4}$$

$$= \{\boldsymbol{x}_i, \{\hat{\boldsymbol{y}}_i^j : \hat{\boldsymbol{y}}_i^j \sim \mathbf{M}_p(\boldsymbol{x}_i)\}_{j=1}^m\}_{i=1}^{N_u}. \tag{5}$$

Here, $\mathbf{M}_p(\boldsymbol{x})$ generates a caption based the content of the image $\boldsymbol{x}$.

4. In turn, we can train the model $\mathbf{M}$ on the unpaired dataset supplemented with synthetic captions, employing the same objective and procedure as in Eq. 3 and Step 2, resulting in a model $\mathbf{M}_u$. Notably, we observed that training solely on synthetic pairs tends to yield small loss values, particularly when resuming from a checkpoint utilized for caption generation. Consequently, we opt to train the model from scratch in this phase to avoid overfitting.

5. Finally, we utilize $\mathbf{M}_u$ to synthesize a new set of captions for the paired data, generating $m$ captions for each image in $\mathcal{D}_p$ following Eq. 4. Once the synthetic captions are generated, we prompt the LLM $\mathbf{G}$ to merge the information of the synthetic caption $\hat{\boldsymbol{y}}_s$ and original caption $\hat{\boldsymbol{y}}_o$, as:

$$\tilde{\boldsymbol{y}}_i = \mathbf{G}(\hat{\boldsymbol{y}}_o, \hat{\boldsymbol{y}}_s),\ \forall \boldsymbol{x}_i \tag{6}$$

$$\hat{\boldsymbol{y}}_o \sim \mathbf{G}(\boldsymbol{y}_i), \hat{\boldsymbol{y}}_s \sim \mathbf{M}_u(\boldsymbol{x}_i), \tag{7}$$

resulting in an augmented paired data $\mathcal{E}(\mathbf{D}_p;\mathbf{M},\mathbf{G}) = \{\boldsymbol{x}_i, \{\tilde{\boldsymbol{y}}_i^j\}_{j=1}^m\}_{i=1}^{N_p}$. As LIST approaches the end of one cycle, we have the option to either conclude the iteration or return to Step 2, thereby allowing the process to continue in a loop.

Overall, LIST functions in a dynamic, synergistic loop, alternating between model training (Steps 2 and 4) and data synthesis (Steps 1, 3, and 5), each phase complementing and enhancing the other. In the following two subsections, we will delve into the details of these two components.

## 3.2 MODEL AND TRAINING

**Model architecture.** LIST is a generic framework and is agnostic to the specific architecture of the vision-language model $\mathbf{M}$. Throughout this paper, we opt for the Contrastive Captioner (CoCa) (Yu et al., 2022) to instantiate $\mathbf{M}$ because of its simplicity and its capability to generate descriptive captions for vision-language learning (*c.r.* (Nguyen et al., 2023)). $\mathbf{M}$ compromises of three components (depicted in Fig. 4 of Appendix).

- A *vision encoder* $\mathbf{E}_v$ instantiated by a Vision Transformer (Dosovitskiy et al., 2020). It takes an image $\boldsymbol{x}$ as input, outputs a global embedding $\boldsymbol{v}_g$ and an array of local embeddings $\boldsymbol{V}_l$.

- A *language encoder* $\mathbf{E_t}$ instantiated by a bidirectional transformer (Vaswani et al., 2017), producing a global embedding $\boldsymbol{t}_g$ for a given caption $\boldsymbol{y}$.

- A *language decoder* $\mathbf{D}_t$ that is instantiated by a unidirectional transformer (Vaswani et al., 2017). It processes the input caption $\boldsymbol{y}$ with the causal masking scheme and conditions on the vision embedding $\boldsymbol{V}_l$. $\mathbf{D}_t$ is tasked to predict the next in the sequence, ultimately outputting a score $\boldsymbol{s}$.

In our empirical observations, initiating training of the model $\mathbf{M}$ from scratch often led to severe overfitting, a phenomenon especially marked in scenarios with limited initial paired data, such as only a few hundred thousand pairs in our case. However, the extensive availability of high-quality unpaired unimodal data provides a beneficial alternative. This plentiful resource enables us to employ unimodal pretrained models as an effective countermeasure against overfitting. Our approach aligns with the LiT methodology (Zhai et al., 2022), in which we opt for a pretrained, frozen vision encoder, specifically DINOv2 (Oquab et al., 2023). This is complemented by a randomly initialized, trainable attentional pooling layer (Yu et al., 2022) atop the pretrained encoder, forming our standard configuration. In the event of the audio-language setting, we simply replace the vision encoder with an audio encoder pretrained by AudioMAE (Huang et al., 2022b) and keep the other audio-irrelevant designs unchanged. Furthermore, considering the language modeling aspect of our model, we initiate the language segments $\mathbf{E}_t, \mathbf{D}_t$ of $\mathbf{M}$ with a pretrained T5 encoder-decoder (Raffel et al., 2020) and use an averaging pooling when extracting the global language embedding with the language encoder. Subsequently, we update the weights of these segments through gradient descent, thereby fine-tuning the model for the tasks of captioning and contrastive learning.

**Training objective.** Following CoCa (Yu et al., 2022), we train the model $\mathbf{M}$ jointly with the contrastive loss $\mathcal{L}_{\text{con}}$ and the caption loss $\mathcal{L}_{\text{cap}}$, weighted by two hyper parameter $\alpha, \beta$ as:

$$\mathcal{L}(\boldsymbol{x}, \boldsymbol{y}; \mathbf{M}) = \alpha * \mathcal{L}_{\text{con}}(\boldsymbol{x}, \boldsymbol{y}) + \beta * \mathcal{L}_{\text{cap}}(\boldsymbol{x}, \boldsymbol{y}). \tag{8}$$

Specifically, the vision encoder $\mathbf{E}_v$ and language encoder $\mathbf{E}_t$ are optimized by the contrastive loss:

$$\mathcal{L}_{\text{con}}(\boldsymbol{x}, \boldsymbol{y}) = -\sum_{i=1}^{N} \log \frac{\exp(\text{sim}(\boldsymbol{v}_g^i, \boldsymbol{t}_g^i)/\tau)}{\sum_{j=1}^{N} \exp(\text{sim}(\boldsymbol{v}_g^i, \boldsymbol{t}_g^j)/\tau)} - \sum_{i=1}^{N} \log \frac{\exp(\text{sim}(\boldsymbol{t}_g^i, \boldsymbol{v}_g^i)/\tau)}{\sum_{j=1}^{N} \exp(\text{sim}(\boldsymbol{t}_g^i, \boldsymbol{v}_g^j)/\tau)}, \tag{9}$$

where the first term accounts for the image-to-text contrastive loss while the second accounts for the text-to-image one, $\text{sim}(\cdot)$ denote the cosine similarity, $\tau$ is a temperature parameter scaling the logits, and $N$ is the batch size.

The vision encoder $\mathbf{E}_v$ and language decoder $\mathbf{D}_t$ are optimized by the caption loss in an autoregressive manner:

$$\mathcal{L}_{\text{cap}}(\boldsymbol{x}, \boldsymbol{y}) = -\sum_{i=1}^{N} \sum_{t=1}^{T_i} \log p(\boldsymbol{y}_t^i | \boldsymbol{y}_1^i, ..., \boldsymbol{y}_{t-1}^i; \boldsymbol{V}_l^i), \tag{10}$$

where $T_i$ is the length of the caption $\boldsymbol{y}^i$, $\boldsymbol{y}_j^i$ is the $j$-th word in $\boldsymbol{y}^i$. $p(\boldsymbol{y}_t^i | \boldsymbol{y}_1^i, ..., \boldsymbol{y}_{t-1}^i; \boldsymbol{V}_l^i)$ is the probability of the $t$-th word in the caption, conditioned on the vision local embedding $\boldsymbol{V}_l^i$ and all the previous words in the caption.

### 3.3 DATA ENGINE

**Captions synthesis.** Utilizing our trained model $\mathbf{M}$, we can generate $m$ captions for a given input image $\boldsymbol{x}$ through standard autoregressive decoding, defined as:

$$\tilde{\boldsymbol{y}} = \arg\max_{\tilde{\boldsymbol{y}}} \prod_{t=1}^{T} P(\tilde{\boldsymbol{y}}_t | \tilde{\boldsymbol{y}}_1, \tilde{\boldsymbol{y}}_2, ..., \tilde{\boldsymbol{y}}_{t-1}; \boldsymbol{V}_l). \tag{11}$$

This decoding process is terminated either when $t \geq T$ or upon sampling an "end of sequence" token. Following this, we apply a standard deduplication procedure for the generated text data. We

Figure 2: **Caption refinement** for paired data. We employ off-the-shelf LLMs, such as LLaMa 2 (Touvron et al., 2023), instructing them to generate a new caption by merging the web-scraped caption with the one synthesized by the captioner. To facilitate in-context learning, we provide the LLM with the task description and several illustrative examples.

utilize MinHash (Broder, 1997; Mou et al., 2023) to eliminate captions that are less than five tokens in length and discard those exhibiting a Jaccard similarity greater than 0.7.

**Captions refinement.** The data engine $\mathcal{E}$ augments the existing captions for the paired data with the LLM $\mathbf{G}$, specifically a LLaMa-2-7B, independent of the presence of the captioner $\mathbf{M}$. In scenarios where $\mathbf{M}$ is not yet integrated, *i.e.*, at the beginning of the LIST loop, our refinement method reverts to the Language Rewrite method (Fan et al., 2023). Here, the LLM $\mathbf{G}$ receives instructions to "rewrite the caption differently", supplemented by several in-context examples that have been previously rewritten by chatbots, such as ChatGPT (OpenAI, 2023), or by humans.

The captioner model $\mathbf{M}$ has the capability to supplement the existing knowledge found in web-scraped captions with novel insights. As extensively analyzed by Nguyen et al. (2023), synthetic captions, as opposed to raw, web-scraped ones, exhibit distinct characteristics. Synthetic captions typically demonstrate greater consistency and coherence with the visual content but lack diversity. On the other hand, raw captions, while offering semantically richer context, are often susceptible to noise, a common byproduct of web-scraping processes. To harness the strengths of both types, we propose instructing LLMs to adeptly integrate the valuable elements from each, thus creating more comprehensive and enriched captions.

Our process begins with the collection of a few integration examples using ChatGPT (OpenAI, 2023), a more capable LLM. We commence by randomly selecting 20 captions from $\mathcal{E}(\mathcal{D}_p; \mathbf{G})$ along with their corresponding synthetic captions. For each pair, we generate a merged caption using a prompt like "Combine a web-scraped caption with a synthesized one, giving precedence to the former." These merged samples then serve as in-context examples. Coupled with the task description, "*From a web-scraped caption || a synthesized caption, create a new caption after ->, favoring the web-scraped details and carefully adding from the synthesized one*", and the specific query, they are used to prompt the relatively smaller LLM $\mathbf{G}$, LLaMa 2 (Touvron et al., 2023) in our case, to integrate the captions. An illustration of this prompting technique is provided in Fig. 2.

## 4 EXPERIMENTS

### 4.1 EXPERIMENT SETUP

**Pretraining data.** Our primary experiments are conducted using the CC3M dataset (Sharma et al., 2018), which comprises 3.3 million high-quality image-text pairs sourced from the web and subsequently subjected to rigorous automatic and manual cleaning processes. We utilize the img2dataset toolbox (Beaumont, 2021) to download the dataset using the provided URL-caption pairs, ultimately yielding approximately 2.8 million image-text pairs, a reduction primarily attributable to expired links. For the audio experiments, we choose the widely used and high-quality audio-caption datasets as the data source. Concretely, we adopt the following configurations of the **paired dataset**:

- **CC160K**: This subset, comprising roughly 5% of the total data pairs in CC3M, is created using the first 160K URL-caption pairs from the dataset.
- **CC600K**: This configuration employs the first 600K URL-caption pairs, amounting to about 20% of the CC3M dataset's total data pairs.

Table 1: **Zero-shot classification accuracy (%) on the ImageNet-1K validation set**. The number in braces denotes the performance gain of LIST compared to CoCa.

| METHOD | NUM. OF DATA | | IMAGENET | |
|---|---|---|---|---|
| | Paired | Unpaired | Top1 | Top5 |
| CLIP (Radford et al., 2021) | 2.8M | 0 | 40.25 | 62.27 |
| CoCa (Yu et al., 2022) | 2.8M | 0 | 39.94 | 62.18 |
| CLIP (Radford et al., 2021) | 160K | 0 | 29.99 | 55.80 |
| LaCLIP (Fan et al., 2023) | 160K | 0 | 32.83 | 58.67 |
| CoCa (Yu et al., 2022) | 160K | 0 | 29.98 | 55.61 |
| LIST | 160K | 2.6M | 36.03 (+6.1) | 59.05 (+3.4) |
| CLIP (Radford et al., 2021) | 600K | 0 | 33.68 | 58.88 |
| LaCLIP (Fan et al., 2023) | 600K | 0 | 39.58 | 63.91 |
| CoCa (Yu et al., 2022) | 600K | 0 | 33.71 | 58.70 |
| LIST | 600K | 2.2M | 41.01 (+7.3) | 64.29 (+5.6) |

Table 2: **Zero-shot classification accuracy on the VTAB benchmark.** The number in braces indicates the performance improvement achieved by LIST over CoCa. We have omitted the **baseline results for CC160K**, as they **did not surpass** the performance level of **random guesses.**

| METHOD | # DATA | | NATURAL | | | | | | SPECIALIZED | | AVG. |
|---|---|---|---|---|---|---|---|---|---|---|---|
| | Paired | Unpaired | Caltech101 | CIFAR100 | DTD | Flowers102 | Pets | SVHN | EuroSAT | RESISC45 | |
| CLIP (Radford et al., 2021) | 2.8M | 0 | 74.38 | 72.13 | 62.93 | 21.58 | 14.80 | 7.60 | 29.57 | 31.12 | 39.26 |
| CoCa (Yu et al., 2022) | 2.8M | 0 | 73.93 | 71.88 | 35.48 | 25.92 | 15.94 | 10.99 | 30.57 | 32.52 | 37.15 |
| LIST | 160K | 2.6M | 67.16 | 58.45 | 21.38 | 11.81 | 9.95 | 9.82 | 23.61 | 26.92 | 28.64 |
| CLIP (Radford et al., 2021) | 600K | 0 | 72.93 | 69.97 | 33.56 | 15.37 | 12.05 | 7.49 | 41.35 | 27.92 | 35.08 |
| LaCLIP (Fan et al., 2023) | 600K | 0 | 75.79 | 73.83 | 36.12 | 18.98 | 15.21 | 13.18 | 39.39 | 32.51 | 38.13 |
| CoCa (Yu et al., 2022) | 600K | 0 | 68.92 | 61.85 | 23.94 | 11.89 | 9.62 | 7.59 | 32.00 | 24.40 | 30.03 |
| LIST | 600K | 2.2M | 74.98 | 75.50 | 40.16 | 20.49 | 15.70 | 18.35 | 40.00 | 30.51 | 39.46 |
| (Gain over CoCa) | | | (+5.1) | (+13.6) | (+16.2) | (+8.6 ) | (+6.1) | (+10.8) | (+8.0) | (+6.1) | (+9.4) |

- **AudioCaps+Clotho**: This configuration combines the 49K pairs from AudioCaps (Kim et al., 2019) dataset and another 4K pairs from the Clotho (Drossos et al., 2020) dataset.

In the first two configurations, we treat the remaining images in the CC3M dataset as **unpaired data**, discarding all associated captions to maintain consistency in our experimental setup. We default to CC160K in all ablation studies unless otherwise specified. For the process of data synthesis, we default to generating five augmented captions, setting $m = 5$. For the audio-language experiments, we use a subset of 730K audio files downloaded according to AudioSet (Gemmeke et al., 2017) as unpaired data. In the caption refinement stage, we opt for the 7B version of LLaMa 2 (Touvron et al., 2023), chosen for its proven efficacy and efficiency.

**Pretraining configurations.** We perform all our experiments on the OpenCLIP codebase (Ilharco et al., 2021), with all the training details given in Sec. B[1].

**Zero-shot evaluation.** We implement the zero-shot evaluation methodology as described in the original CLIP paper (Radford et al., 2021), focusing primarily on the top1/top5 accuracy on ImageNet validation set (Deng et al., 2009) to assess performance. This process involves utilizing 80 prompt templates, for instance, 'a photo of {object}', to calculate the average text embedding for each class, serving as the classifier. Subsequently, each image is classified based on the proximity between its global embedding and these averaged text classifiers, effectively leveraging the learned associations between images and textual descriptions. Similar procedures go for audio classification.

---

[1]Code will be made publicly available along with the paper.

Table 3: **Zero-shot image-text retrieval** on MS-COCO, measured by the Recall@K (K=1, 5, 10).

| METHOD | # DATA | | TEXT-TO-IMAGE | | | IMAGE-TO-TEXT | | |
|---|---|---|---|---|---|---|---|---|
| | Paired | Unpaired | R@1 | R@5 | R@10 | R@1 | R@5 | R@10 |
| CLIP (Radford et al., 2021) | 2.8M | 0 | 28.69 | 54.11 | 65.99 | 37.92 | 65.48 | 76.38 |
| CoCa (Yu et al., 2022) | 2.8M | 0 | 29.95 | 55.39 | 66.38 | 39.40 | 67.14 | 77.28 |
| CLIP (Radford et al., 2021) | 600K | 0 | 24.70 | 50.38 | 62.26 | 33.92 | 61.82 | 73.22 |
| LaCLIP (Fan et al., 2023) | 600K | 0 | 25.93 | 51.04 | 62.26 | 36.78 | 64.14 | 75.18 |
| CoCa (Yu et al., 2022) | 600K | 0 | 25.22 | 50.42 | 62.48 | 34.44 | 62.06 | 73.04 |
| LIST | 600K | 2.2M | 29.66 | 54.81 | 66.09 | 41.56 | 68.34 | 78.76 |
| (Gain over CoCa) | | | (+4.44) | (+4.39) | (+3.61) | (+7.12) | (+6.28) | (+5.72) |

## 4.2 ZERO-SHOT IMAGE CLASSIFICATION

**ImageNet.** ImageNet (Deng et al., 2009) is one of the golden benchmarks for assessing vision-related models. The results of our experiments are summarized in Tab. 1, where the significant performance improvements achieved by LIST are evident. Specifically, LIST surpasses CoCa by absolute margins of 6.2% and 7.3% in top-1 zero-shot classification accuracy, utilizing 160K and 600K pairs for training, respectively. Additionally, it is noteworthy that LIST, when trained with 600K data pairs, outperforms baseline models trained with 2.8M pairs – approximately four times the data used for LIST.

**VTAB.** We further assess the performance of LIST using the VTAB benchmark (Zhai et al., 2019), which is a comprehensive collection of datasets designed to evaluate models from multiple perspectives. Our experiments, conducted on the CLIPBenchmark codebase (LAION, 2022), focus on a subset of VTAB that includes realistic image datasets: the natural sets featuring Caltech101 (Fei-Fei et al., 2004), CIFAR100 (Krizhevsky, 2009), DTD (Cimpoi et al., 2014), Flowers102 (Nilsback & Zisserman, 2008), Pet (Parkhi et al., 2012), and SVHN (Netzer et al., 2011); and the specialized set with EuroSAT (Helber et al., 2017) and RESISC45 (Cheng et al., 2017). The results, detailed in Tab. 2, reveal that LIST, trained with 600K paired data, consistently surpasses CoCa across all categories, with an average improvement of 9.4% in absolute terms. Notably, LIST also slightly outperforms CLIP/CoCa models trained with a larger dataset of 2.8M pairs, underscoring its capability of leveraging the abundant unpaired data. Note that, we omit the results for the baselines when trained on CC160K because they perform at random-guess level. We conjecture this issue might be a result of the large distribution shift from the training data to the test data, and LIST can significantly bypass it thanks to the leverage of unpaired data and diverse synthetic captions.

## 4.3 ZERO-SHOT IMAGE-TEXT RETRIEVAL

We evaluate LIST on the MS-COCO image-text retrieval task (Lin et al., 2014). We directly follow the setting in CLIPBenchmark (LAION, 2022) and report the results on both image-to-text retrieval and text-to-image retrieval in Tab. 3. On both tasks and in terms of all metrics, we can see that, even compared to the baselines using $20\times$ more pairs, LIST consistently outperforms them by a large margin, *e.g.*, as large as 7% compared with CoCa, suggesting the generalizability of LIST to tasks other than classification.

## 4.4 ZERO-SHOT AUDIO CLASSIFICATION

We extend the application of LIST to audio-language alignment, a challenging scenario typically constrained by the availability of only a few thousand paired examples, such as the 53K pairs in combination from AudioCaps (Kim et al., 2019) and Clotho (Drossos et al., 2020) datasets, and 730K unpaired audios from AudioSet (Gemmeke et al., 2017) dataset. Without any twists-and-bells, we observe that LIST obtains 43.9% 0-shot audio classification accuracy on the ESC-50 (Piczak, 2015) dataset, outperforming CoCa by 2%. This demonstrates the capability of LIST to generalize beyond vision-language tasks to the alignment of other modalities, even in a small-data regime.

Table 4: **Zero-shot classification accuracy (%) on the ESC-50 validation set**. The number in braces denotes the performance gain of LIST compared to CoCa.

| METHOD | NUM. OF DATA | | ESC-50 ZERO-SHOT ACCURACY | |
|---|---|---|---|---|
| | Paired | Unpaired | Top1 | Top5 |
| CLAP (Elizalde et al., 2023) | 53K | 0 | 41.5 | 74.1 |
| CoCa (Yu et al., 2022) | 53K | 0 | 42.1 | 75.2 |
| LIST | 53K | 730K | 43.9 (+1.8) | 76.6(+1.4) |

Table 5: **Compositionality evaluation on SugarCrepe.** The number in braces indicates the performance improvement achieved by LIST over CoCa. The table shows that LIST, trained with only 160K paired data, even surpasses CoCa trained $\sim 20\times$ larger dataset of 2.8 million pairs in 4 out of 7 entries, and matches it in one.

| METHOD | # DATA | | REPLACE | | | SWAP | | ADD | |
|---|---|---|---|---|---|---|---|---|---|
| | Paired | Unpaired | Obj. | Attr. | Rel. | Obj. | Attr. | Obj. | Attr. |
| CLIP (Radford et al., 2021) | 2.8M | 0 | 88.98 | 71.45 | 66.29 | 59.35 | 57.68 | 75.27 | 68.21 |
| CoCa (Yu et al., 2022) | 2.8M | 0 | 90.07 | 73.73 | 69.20 | 53.25 | 59.46 | 76.43 | 68.50 |
| CLIP (Radford et al., 2021) | 160K | 0 | 86.08 | 66.75 | 54.94 | 59.35 | 51.50 | 73.13 | 69.94 |
| LaCLIP (Fan et al., 2023) | 160K | 0 | 89.23 | 65.99 | 57.18 | 60.57 | 56.01 | 77.21 | 70.81 |
| CoCa (Yu et al., 2022) | 160K | 0 | 85.64 | 66.12 | 55.41 | 56.50 | 51.95 | 74.36 | 66.91 |
| LIST | 160K | 2.6M | 90.13 | 72.72 | 65.29 | 62.60 | 59.46 | 79.15 | 73.12 |
| (Gain over CoCa) | | | (+4.5) | (+6.5) | (+9.9) | (+6.1) | (+7.5) | (+4.8) | (+6.2) |
| CLIP (Radford et al., 2021) | 600K | 0 | 88.86 | 70.81 | 63.58 | 59.76 | 55.41 | 76.82 | 67.92 |
| LaCLIP (Fan et al., 2023) | 600K | 0 | 89.89 | 71.45 | 61.02 | 58.13 | 60.51 | 78.78 | 67.63 |
| CoCa (Yu et al., 2022) | 600K | 0 | 87.71 | 69.53 | 63.51 | 58.13 | 56.46 | 76.92 | 68.50 |
| LIST | 600K | 2.2M | 90.86 | 75.51 | 66.36 | 64.23 | 61.26 | 79.78 | 78.18 |
| (Gain over CoCa) | | | (+3.2) | (+6.0) | (+2.9) | (+6.1) | (+4.8) | (+2.9) | (+9.7) |

## 4.5 COMPOSITIONALITY EVALUATION

Recent studies (Thrush et al., 2022; Yuksekgonul et al., 2022; Hsieh et al., 2023) have raised questions about the compositional capabilities of vision-language models, uncovering a tendency for these models to behave more like bag-of-words systems rather than fully understanding attributes, relationships, and the order of objects. In this context, we explore the compositional understanding of LIST using the SugarCrepe dataset (Hsieh et al., 2023), which is carefully crafted by altering captions through objects/attributes replacements, swaps, and additions and relations replacements. As shown in Tab. 5, LIST is compared with baseline models under various configurations, demonstrating that our method significantly outperforms the baselines with an equivalent amount of paired data. Notably, LIST, even when trained with only 160K paired data, surpasses CoCa trained with a much ($\sim 20\times$) larger dataset of 2.8 million pairs in 4 out of 7 entries, and matches it in one. This highlights the exceptional ability of LIST to effectively utilize unpaired data, thereby enhancing the compositional understanding in vision-language models.

## 4.6 ANALYSES

**Scale LIST along data-/model-axes.** We conduct experiments with 600K/1.5M data pairs from CC3M and 3M/10M unpaired images from LAION400M (Schuhmann et al., 2022), with results detailed in Fig. 3a. We observe a promising scaling behavior of LIST w.r.t. data size: i) when the number of paired data is fixed, LIST's performance monotonously improves as the quantity of unpaired images increases; and ii) when the number of unpaired images is fixed, LIST enjoys a consistent performance boost with more paired data. Moreover, despite the different sources of paired and unpaired data, LIST significantly outperforms baselines using only paired data, a gain further amplified when mixing data distributions from both sources (see Sec. C). In Fig. 3b, our evaluation of LIST's scalability using the ViT-L/T5-L model shows a significant performance improvement, mirroring the trends seen in CoCa's performance. The above experiments underscore LIST's ability

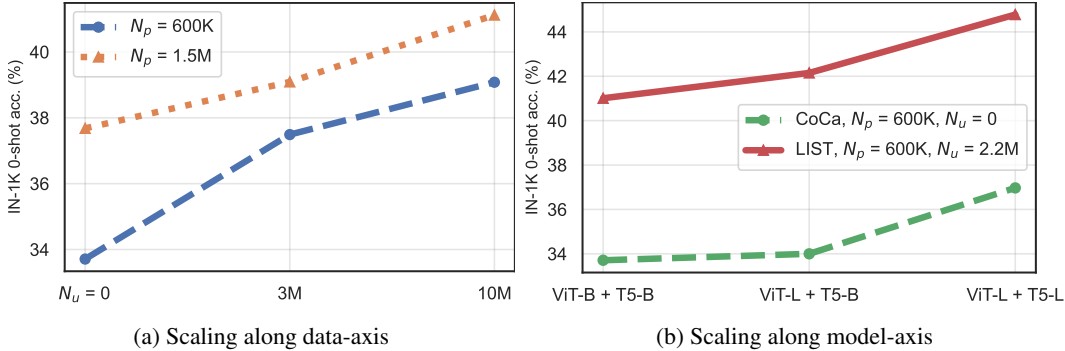

Figure 3: **Scaling LIST along data- and model-axes**. Sources of paired and unpaired data: a) CC3M and LAION400M; and b) CC3M as in Sec. 4.1.

Table 6: **Impact of caption refinement.** The synthesis of captions for paired data followed by their merge with original captions using LLMs yields the best results.

| DATA | $\mathcal{D}_p$ | $\mathcal{E}(\mathcal{D}_p)$ IN STEP 2 | $\mathcal{E}(\mathcal{D}_p)$ IN STEP 5 | |
|---|---|---|---|---|
| MERGE | × | × | × | ✓ |
| $\mathbf{M}_p$ | 28.36 | 30.44 | 33.16 | 34.92 |

Table 7: **Number of training loops in LIST.** Model performance progressively improves with each training loop, but tends to plateau at the second iteration.

| #CYCLES | 0 | 1 | | 2 | |
|---|---|---|---|---|---|
| MODEL | $\mathbf{M}_p$ | $\mathbf{M}_u$ | $\mathbf{M}_p$ | $\mathbf{M}_u$ | $\mathbf{M}_u$ |
| ImageNet | 30.44 | 36.03 | 34.92 | 36.21 | 34.98 |

to scale with the data and model size and its capability to cope with distribution discrepancy between paired and unpaired data, highlighting its potential in more practical scenarios.

**Caption refinement.** We conducted a series of experiments to evaluate the effectiveness of our caption refinement approach, which leverages LLMs. The results are comprehensively summarized in Tab. 6. Our observations indicate that while the text-only caption augmentation in Step 2 of Tab. 3.1 significantly enhances the performance compared to the baseline $\mathcal{D}_p$, the process of generating captions for paired data using the trained vision-language model and subsequently merging them with original captions through LLMs leads to even further improvements. This set of experiments highlights the importance of generating captions based on visual cues, the synergy between synthetic and original captions, and the advantages of employing LLMs to seamlessly integrate these diverse information sources.

**Number of training cycles.** Tab. 7 provides a summary of our experiments exploring the impact of the number of LIST loops on model performance. It is observed that the model's performance improves progressively with respect to each training loop. However, this enhancement appears to plateau after the second iteration. Consequently, for the sake of efficiency, we have chosen to limit LIST to a single loop as the default setting.

## 5 CONCLUSION

This paper introduces the Language-Image Self-Training (LIST) framework, a novel methodology for multi-modal alignment. LIST's distinctive approach, which adeptly leverages the untapped potential of unpaired data, mitigates the traditional reliance on large-scale, paired multi-modal datasets. This framework, characterized by its synergistic cycle of model training and data synthesis, and further enhanced by the integration of Large Language Models, significantly improves both data quality and model performance. Comprehensive evaluations on a wide range of standard zero-shot classification, retrieval, and compositionality benchmarks not only demonstrate LIST's effectiveness in enhancing vision-language alignment but also highlight its capability to generalize to new modalities, *e.g.*, audio-language.

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

Table 8: **Linear probing on VTAB** (Zhai et al., 2019) & **robustness evaluation** on several variants of ImageNet benchmark, ImageNetv2 (Shankar et al., 2020), ImageNet-A (Hendrycks et al., 2021b), and ImageNet-R (Hendrycks et al., 2021a).

| METHOD | LINEAR PROBING ON VTAB | | ZERO-SHOT CLASSIFICATION | | |
|---|---|---|---|---|---|
| | Natural | Specialized | INv2 | IN-A | IN-R |
| CoCa | 84.07 | 92.80 | 34.09 | 29.33 | 49.11 |
| LIST | 85.54 (+1.5) | 93.58 (+0.8) | 36.08 (+2.0) | 30.41 (+1.1) | 53.69 (+4.5) |

Table 9: **Influence of the number of augmented captions** $m$. Within each block, $\mathbf{M}_u$ is trained using synthetic data produced by $\mathbf{M}_p$.

| MODEL | $m$ | IMAGENET | |
|---|---|---|---|
| | | Top1 | Top5 |
| $\mathbf{M}_p$ | 1 | 28.36 | 51.28 |
| $\mathbf{M}_u$ | 1 | 32.27 | 57.98 |
| $\mathbf{M}_u$ | 5 | 33.52 | 58.85 |
| $\mathbf{M}_p$ | 5 | 30.44 | 53.02 |
| $\mathbf{M}_u$ | 5 | 36.03 | 59.05 |

Table 10: **Impact of pretrained vision & language components.** While pretrained models are advantageous on their own, their combination with self-training significantly enhances overall efficacy.

| MODEL | PRETRAINED | | IMAGENET | |
|---|---|---|---|---|
| | Vision | Language | Top1 | Top5 |
| $\mathbf{M}_p$ | × | × | 2.27 | 7.01 |
| $\mathbf{M}_u$ | × | × | 3.14 | 8.23 |
| $\mathbf{M}_p$ | ✓ | × | 22.90 | 40.45 |
| $\mathbf{M}_p$ | ✓ | ✓ | 30.44 | 53.02 |
| $\mathbf{M}_u$ | ✓ | ✓ | 36.03 | 59.05 |

## A  ADDITIONAL ANALYSES

**Linear probing & robustness evaluation.** In Tab. 8, we report the averaged linear probing results over 5 natural datasets and 2 specialized datasets of VTAB (Zhai et al., 2019) and assess LIST's robustness on three ImageNets variants with image/label distribution shifts, namely ImageNetv2 (Shankar et al., 2020), ImageNet-A (Hendrycks et al., 2021b), and ImageNet-R (Hendrycks et al., 2021a). We can see that LIST performs consistently better than CoCa on all entries, verifying its efficacy under various settings and robustness to distribution shifts.

**Number of augmented captions** $m$. Tab. 9 presents a detailed analysis of how varying the number of captions $m$ impacts different stages of the LIST framework. Within each block of the table, $\mathbf{M}_u$ is trained using synthetic data produced by $\mathbf{M}_p$. A key observation is that training with multiple captions benefits both $\mathbf{M}_p$ (compare 1st and 4th rows) and $\mathbf{M}_u$ (compare 2nd and 3rd rows). Furthermore, this approach also enhances the quality of the generated captions. This improvement is particularly evident when comparing the 3rd row with the 5th row, where the models employ captions generated by $\mathbf{M}_p$ trained with just one caption (1st row) and five captions (4th row), respectively.

**Pretrained image/text components.** To discern the contributions of the self-training framework versus the utilization of pretrained image encoders and text encoder-decoders, we carried out comparative experiments involving models with and without these components. The results, as shown in Tab. 10, reveal that the integration of more pretrained components consistently and significantly enhances model performance when trained on paired data (as evidenced by comparing the 1st, 3rd, and 4th rows) and also improves the quality of the generated captions (noted in the comparison between the 2nd and 5th rows). Furthermore, our findings show that self-training, independent of the use of pretrained models, is adept at extracting valuable information from unpaired data, as demonstrated by the decent performance of $\mathbf{M}_u$ trained solely on unpaired data. The addition of pretrained models further amplifies the efficacy of self-training, unlocking its full potential.

**Comparison to SLIP and SigLIP.** As displayed in Tab. 11, we can see that applying extra self-supervision with the unpaired data (SLIP (Mu et al., 2022)) only brings marginal gain, partly because all models here already use SSL pretrained model. SigCLIP (Zhai et al., 2023) offers considerable gains over CLIP but still largely underperforms LIST. Moreover, we note that these explorations

Table 11: **Zero-shot image classification on IN-1K**, with 600K pairs available for each methods.

| METHOD | CLIP | SLIP | SigLIP | LIST | LIST + SigLIP |
|---|---|---|---|---|---|
| TOP1 ACC. | 33.68 | 33.82 | 36.40 | **41.01** | **42.09** |

Table 12: **Zero-shot image classification on IN-1K**, with 600K pairs available for each methods.

| CAPTION SOURCE | Raw | Synthetic (LLaVA) | Synthetic (Ours) |
|---|---|---|---|
| 0-shot acc. | 29.98 | 27.82 | **31.82** |

are orthogonal to ours: incorporating SigLIP with LIST can further improve the model performance by 1% in absolute. These experiments demonstrate the effectiveness of the self-training scheme in LIST and indicate further improvements of LIST by adopting better loss functions.

**Comparison to simply using Multi-Modal LLMs for captions generation.** Existing MLLMs, such as LLaVA (Liu et al., 2023), are trained using a mixture of common data sources (e.g., CC3M) and often include components exposed to billions of data pairs (e.g., the CLIP encoder). Using them to refine captions might hinder the effort to isolate the effective contributions of self-training. Nonetheless, we observed that captions generated by LLaVA lack diversity. Training on these captions results in lower loss but also lower accuracy than using raw captions (see Tab. 12, all entries with one caption per image).

## B TRAINING CONFIGURATIONS

We perform all our experiments on the OpenCLIP codebase (Ilharco et al., 2021) with PyTorch 2.0 (Paszke et al., 2019) and the automatic mixed precision training. The input image undergoes a weak augmentation, *i.e.*, random flip, random crop, and is then resized to $224 \times 224$. The input text is tokenized by a SentencePiece tokenizer (Kudo & Richardson, 2018), with a maximal length of 40 tokens to avoid computation burden. In our experiments, we use the base-size Transformers (Vaswani et al., 2017), *i.e.*, ViT-Base/14 (Dosovitskiy et al., 2020) pretrained by DINOv2 (Oquab et al., 2023) for the vision encoder $\mathbf{E}_v$ and T5-Base (Raffel et al., 2020) for the language encoder-decoder $\mathbf{E}_t, \mathbf{D}_t$. The model is trained using the AdamW (Loshchilov & Hutter, 2017) optimizer, with a batch size of 2,048 for both images and texts, a weight decay set to 0.2, an initial $\tau$ set to $1/0.07$ (Ilharco et al., 2021), and the cosine annealing learning rate decay (Loshchilov & Hutter, 2016). We keep the image encoder fixed, except for the attentional pooling layer. The hyperparameters $\alpha$ and $\beta$ in Eq. 8 are set to 1 and 2, respectively, following (Yu et al., 2022).

During the training process, we observed that the language decoder $\mathbf{D}_t$ necessitates a larger gradient update step compared to the language encoder. In light of this, specifically for effective caption synthesis, we opt to train the model for 128 epochs with a learning rate of 0.002. Additionally, we adjust the training process by scaling down the gradient of the text encoder by a factor of 0.1. For the evaluation phase, involving both synthetic and original data, the models are trained for a shorter duration of 32 epochs, employing a reduced learning rate of 0.0005, in line with the standard setting of OpenCLIP (Ilharco et al., 2021). To facilitate future research, code and synthesized data will be made publicly available concurrently with the official release of this paper.

## C MIXING PAIRED DATA AND UNPAIRED DATA

Going further than training solely on the augmented real data pairs or the synthetic data pairs, we also explored training on a mixture of both types of data sources. The learning process can be formulated as:

$$\mathbf{M}_{p+u} = \arg\min_{\mathbf{M}} \mathbb{E}_{(\boldsymbol{x},\boldsymbol{y}) \sim \mathcal{D}_m} \mathcal{L}(\boldsymbol{x}, \boldsymbol{y}; \mathbf{M}),$$
$$\text{where } \mathcal{D}_m = w \cdot \mathcal{E}(\mathcal{D}_p) + (1-w) \cdot \mathcal{E}(\mathcal{D}_u). \tag{12}$$

Here, $\mathcal{L}(\cdot)$ is the loss function defined in Eq. 8. $w \in [0,1]$ is a weighting term and controls the proximity of $\mathcal{D}_m$ to $\mathcal{E}(\mathcal{D}_p)$, and thus to $\mathcal{D}_p$. In practice, we implement the weighting by sampling

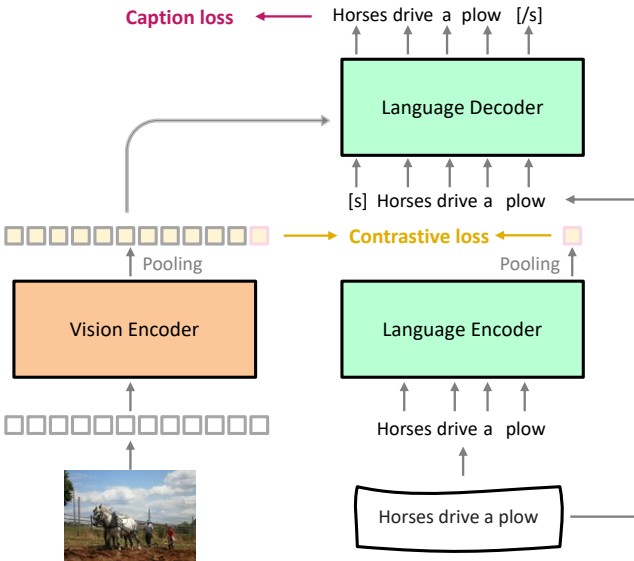

Figure 4: The **captioner model** M compromises 1) a *vision encoder* to encode the image into a global embedding for contrasting and local embeddings for captioning, 2) a bidirectional *language encoder* to encode caption into a global embedding for contrasting, and 3) a unidirectional *language decoder* trained to predict next tokens, conditioned on the vision local embeddings.

Table 13: **Zero-shot classification accuracy (%) on the ImageNet-1K** (Deng et al., 2009) validation set. This table utilizes a frozen ViT-Base encoder pretrained by DINOv2 (Oquab et al., 2023) and a trainable T5-Base encoder-decoder (Raffel et al., 2020) where relevant. The number in braces denotes the performance gain of LIST compared to CoCa.

| METHOD | # DATA | | IMAGENET | |
|---|---|---|---|---|
| | Paired | Unpaired | Top1 | Top5 |
| CLIP (Radford et al., 2021) | 2.8M | 0 | 40.25 | 62.27 |
| CoCa (Yu et al., 2022) | 2.8M | 0 | 39.94 | 62.18 |
| CoCa (Yu et al., 2022) | 160K | 0 | 29.98 | 55.61 |
| LIST | 160K | 2.6M | 36.03 (+6.1) | 59.05 (+3.4) |
| LIST$_{p+u}$ | 160K | 2.6M | 36.99 (+7.0) | 59.69 (+4.1) |
| CoCa (Yu et al., 2022) | 600K | 0 | 33.71 | 58.70 |
| LIST | 600K | 2.2M | 41.01 (+7.3) | 64.29 (+5.6) |
| LIST$_{p+u}$ | 600K | 2.2M | 41.50 (+7.8) | 64.80 (+6.1) |

$w * N$ sample pairs from $\mathcal{E}(\mathcal{D}_p)$ and $(1 - w) * N$ sample pairs from $\mathcal{E}(\mathcal{D}_u)$ for each mini-batch of size $N$.

The models are trained over 64 *paired data epochs*[2] on the CC160K dataset with a weighting factor $w = 0.25$. For the CC600K dataset, training is conducted for 32 paired epochs with $w = 0.5$, a decision influenced by the unpaired data volume being less than four times the volume of paired data. All other experimental settings, including batch size, learning rates, and etc, remain consistent with those described in Sec. 4.1.

Tables 13, 14, and 15 summarize the results of mixing paired and unpaired data. It is evident that merging these two data sources (*i.e.*, LIST$_{p+u}$ as indicated in the tables) typically enhances performance. This improvement is particularly notable in the case of CC160K, where the volume of

---

[2]In this context, 'paired data epochs' refers to the repetition of the paired data 64 times during training. Given the inclusion of samples from unpaired data, the effective number of training epochs is $(1 + \frac{1}{w}) \times$ relative to the paired data epochs.

Table 14: **Zero-shot classification accuracy on the VTAB benchmark** (Zhai et al., 2019). The number in braces indicates the performance improvement achieved by LIST over CoCa. We have omitted the baseline results for CC160K, as they did not surpass the performance level of random guesses.

| METHOD | # DATA | | NATURAL | | | | | | SPECIALIZED | | AVG. |
|---|---|---|---|---|---|---|---|---|---|---|---|
| | Paired | Unpaired | Caltech101 | CIFAR100 | DTD | Flowers102 | Pets | SVHN | EuroSAT | RESISC45 | |
| CLIP (Radford et al., 2021) | 2.8M | 0 | 74.38 | 72.13 | 62.93 | 21.58 | 14.80 | 7.60 | 29.57 | 31.12 | 39.26 |
| CoCa (Yu et al., 2022) | 2.8M | 0 | 73.93 | 71.88 | 35.48 | 25.92 | 15.94 | 10.99 | 30.57 | 32.52 | 37.15 |
| LIST | 160K | 2.6M | 67.16 | 58.45 | 21.38 | 11.81 | 9.95 | 9.82 | 23.61 | 26.92 | 28.64 |
| LIST$_{p+u}$ | 160K | 2.6M | 74.08 | 73.13 | 36.54 | 18.85 | 13.46 | 13.47 | 33.76 | 26.79 | 36.26 |
| CoCa (Yu et al., 2022) | 600K | 0 | 68.92 | 61.85 | 23.94 | 11.89 | 9.62 | 7.59 | 32.00 | 24.40 | 30.03 |
| LIST | 600K | 2.2M | 74.98 | 75.50 | 40.16 | 20.49 | 15.70 | 18.35 | 40.00 | 30.51 | 39.46 |
| (Gain over CoCa) | | | (+5.1) | (+13.6) | (+16.2) | (+8.6 ) | (+6.1) | (+10.8) | (+8.0) | (+6.1) | (+9.4) |
| LIST$_{p+u}$ | 600K | 2.2M | 75.69 | 76.41 | 41.65 | 21.69 | 17.17 | 18.67 | 44.28 | 32.86 | 41.05 |
| (Gain over CoCa) | | | (+6.8) | (+14.6) | (+17.7) | (+9.8) | (+7.6) | (+11.1) | (+12.3) | (+8.4) | (+11.0) |

Table 15: **Compositionality evaluation on the SugarCrepe dataset** (Hsieh et al., 2023). The number in braces indicates the performance improvement achieved by LIST over CoCa.

| METHOD | # DATA | | REPLACE | | | SWAP | | ADD | |
|---|---|---|---|---|---|---|---|---|---|
| | Paired | Unpaired | Obj. | Attr. | Rel. | Obj. | Attr. | Obj. | Attr. |
| CLIP (Radford et al., 2021) | 2.8M | 0 | 88.98 | 71.45 | 66.29 | 59.35 | 57.68 | 75.27 | 68.21 |
| CoCa (Yu et al., 2022) | 2.8M | 0 | 90.07 | 73.73 | 69.20 | 53.25 | 59.46 | 76.43 | 68.50 |
| CoCa (Yu et al., 2022) | 160K | 0 | 85.64 | 66.12 | 55.41 | 56.50 | 51.95 | 74.36 | 66.91 |
| LIST | 160K | 2.6M | 90.13 | 72.72 | 65.29 | 62.60 | 59.46 | 79.15 | 73.12 |
| (Gain over CoCa) | | | (+4.5) | (+6.5) | (+9.9) | (+6.1) | (+7.5) | (+4.8) | (+6.2) |
| LIST$_{p+u}$ | 160K | 2.6M | 90.01 | 73.73 | 66.50 | 61.38 | 59.61 | 78.95 | 73.99 |
| (Gain over CoCa) | | | (+4.4) | (+7.6) | (+10.1) | (+4.9) | (+7.7) | (+4.6) | (+7.1) |
| CoCa (Yu et al., 2022) | 600K | 0 | 87.71 | 69.53 | 63.51 | 58.13 | 56.46 | 76.92 | 68.50 |
| LIST | 600K | 2.2M | 90.86 | 75.51 | 66.36 | 64.23 | 61.26 | 79.78 | 78.18 |
| (Gain over CoCa) | | | (+3.2) | (+6.0) | (+2.9) | (+6.1) | (+4.8) | (+2.9) | (+9.7) |
| LIST$_{p+u}$ | 600K | 2.2M | 91.46 | 73.48 | 66.07 | 64.63 | 65.92 | 80.50 | 73.12 |
| (Gain over CoCa) | | | (+3.8) | (+4.0) | (+2.6) | (+6.5) | (+9.5) | (+3.6) | (+4.6) |

unpaired data is approximately 20 times greater than that of the paired data. However, it's noteworthy that the performance gains on the SugarCrepe dataset are not as significant as those observed in other datasets. This could be due to the models undergoing fewer gradient updates compared to those trained solely on synthetic data while compositional understanding requires more training epochs to differentiate the altered hard-negative captions.

## D LIMITATIONS

A notable limitation of the LIST framework lies in its reliance on a small, initial paired dataset, potentially restricting the diversity of concepts learned and thereby limiting its ability to generalize diverse captions for unpaired data. This reliance might affect the model's scalability and adaptability, particularly in new domains or complex tasks that differ significantly from the initial training data. However, it's important to note that LIST does employ pretrained language encoder-decoders and off-the-shelf LLMs, which were trained on large-scale and diverse unimodal data. This integration enables the introduction of new and unseen concepts into the framework. This aspect of LIST potentially aids in mitigating some of the generalization limitations by infusing a wider range of knowledge and concepts beyond the initial dataset.

# E    BROADER IMPACT

While LIST, as all other works leveraging LLMs, inherits biases from training on the web data, it consistently outperforms baselines using only original captions. This indicates that the diversity from associating $m$ captions per image, along with the use of unpaired data, generally outweigh the risks posed by noisy synthetic captions. Furthermore, the process of image captioning considerably improves the alignment of captions with visual content, evidenced by CLIP Score, thus reducing hallucination. To address other biases, we encourage the use of bias-reduced LLMs and extra filtering before LIST's practical deployment.

