# OpenReview forum: "Self-Training on Unpaired Data Improves Multi-Modal Alignment"
_ICLR.cc/2025/Conference — ICLR 2025 Conference Withdrawn Submission_

### Official Review · Reviewer_Fq5g · 2024-10-17

**Soundness:** 2
**Presentation:** 4
**Contribution:** 2
**Rating:** 3
**Confidence:** 4

**Summary:**

The paper introduces LIST, a novel method for using unpaired images to train image-text models. LIST generates and refines captions for both paired and unpaired images in a continuous cycle to enhance the performance of foundational models. However, the evaluation of the proposed method is insufficient, particularly in comparison with simpler alternatives.

**Strengths:**

1. The paper is clearly written and easy to follow.

1. The proposed LIST technique is novel, using unpaired images by generating or refining captions.

1. Better performance is shown when finetuning image-text models with captions generated by LIST compared to using the raw captions, showing an improvement in benchmark results (e.g., LIST uses 600K paired images and 2.2M generated captions, compared to CoCa’s 2.8M paired samples).

**Weaknesses:**

1. Existing image captioning models, such as Vision-Language models (e.g., ShareGPT4V, LLaVA), already demonstrate excellent performance. Simple solutions like generating captions for unpaired images using these models could suffice, and the need for LIST must be validated by demonstrating significant improvements over these approaches, which the paper does not do.

1. LIST should be evaluated in a setting that uses all paired samples along with unpaired samples to truly assess the necessity of using unpaired data. However, the paper uses a sample of paired examples, leaving this question unaddressed.

1. Most of the baselines in the paper are weak (except the first two rows in Tables 1, 2, 3, and 5), as they involve relatively smaller datasets. For instance, LIST uses 2.6M samples, while the baselines use only 0.6M. The highlighted performance gains are shown against these weaker baselines, making it difficult to assess the real impact of LIST.

1. In the audio-text model experiment, despite using significantly more data compared to the baseline, the performance improvement is not great.

**Questions:**

None

---

### Official Review · Reviewer_C4uy · 2024-10-21

**Soundness:** 3
**Presentation:** 3
**Contribution:** 2
**Rating:** 3
**Confidence:** 4

**Summary:**

The paper proposes a novel method called **L**anguage-**I**mage **S**elf-**T**raining (**LIST**), which leverages multimodal representation alignment, particularly for Vision-Language and Audio-Language paired modalities, even with unpaired datasets. The key idea focuses on addressing the challenges posed by unpaired datasets through two approaches:

- **Captioner Model**: Alternates between training on paired and synthetic data using large language models (LLMs) to enhance data diversity and quality.
- **Data Engine**: Generates diverse captions for both paired and unpaired images, combining synthetic and web-scraped captions to improve training.

For the training process, the paper adopts the training objective functions from CoCa, using contrastive loss to align image-to-text (i2t) and text-to-image (t2i) representations. Overall, the method demonstrates exceptional performance across various benchmark datasets, including zero-shot classification on ImageNet-1K, VTAB, ESC-50, retrieval tasks on MS-COCO, and compositionality evaluation on SugarCrepe, among others.

**Strengths:**

1. **Performance Improvement**: The method achieves significant performance gains across various benchmark datasets, with an average improvement of 5~10%.

2. **Approach with Unpaired Datasets**: Since not all multimodal data has perfectly matched pairs, this presents a non-trivial challenge in real-world applications. This paper addresses the issue by leveraging advanced large language models (LLMs) like ChatGPT and LLaMA to overcome the limitations posed by unpaired datasets.

3. **Extension to Paired Modality**: While recent studies have primarily focused on Vision-Language approaches, this paper extends the research to other paired modalities. It demonstrates performance improvements not only in Vision-Language tasks but also in Audio-Language tasks.

**Weaknesses:**

1. **Reliance on LLM**: This method heavily depends on the reliability of large language models (LLMs). If the LLM struggles to generate accurate captions for image or audio data, the performance could suffer as a result.

2. **Additional Resources for Captioning**: The process of synthesizing captions for both paired and unpaired data adds complexity and overhead, which might not be feasible or scalable for all applications.

3. **Different Semantics Across Modalities**: As highlighted by Huh et al. [1], different modalities can carry distinct semantic meanings. For example, it is challenging to ensure that there is always an appropriate caption for audio data, as the semantics of audio may not always align with those of text.

4. **Limited Novelty**: While this paper demonstrates impressive performance across multiple benchmark datasets, its specific contributions could be further clarified. This is partly due to its reliance on advanced LLMs, which may be considered a more straightforward approach.


[1] Huh, Minyoung, et al. "The platonic representation hypothesis." arXiv preprint arXiv:2405.07987 (2024).

**Questions:**

**Major Question**

1. What happens if the caption does not accurately describe the image or audio data? How does this impact performance?

2. Caption generation remains a key concern. What is the performance difference between using simple captions, e.g., "a photo of a {}", and the synthetic captions generated by the **Captioner Model**? (An additional experiment is not required, but it’s a point of consideration.) How strongly is performance influenced by the accuracy of the captions?

3. In general, multimodal alignment is not limited to one-way directions. For example, this paper only explores one direction, where generating captions (**L**anguage modality) assists the image (**V**ision modality) [**L** $\rightarrow$ **V**] or helping audio (**A**udio modality) [**L** $\rightarrow$ **A**]. If generative models for vision or audio could generate synthetic data for their respective modalities, would it be possible to train in the opposite direction, e.g., [**V $\rightarrow$ L**] or [**A $\rightarrow$ L**]?

**Minor Question**

While the Vision-Language modality is well described, the Audio-Language modality lacks sufficient detail. Could you provide more information on the specifics of this setting?

\
$\Rightarrow$ **If the above questions are thoroughly addressed and clearly explained, I would be happy to consider and more than willing to raising the score.**

---

### Official Review · Reviewer_4yXQ · 2024-11-03

**Soundness:** 2
**Presentation:** 3
**Contribution:** 2
**Rating:** 5
**Confidence:** 4

**Summary:**

Authors propose a data process to leverage off-the-shelf LLM to train a CoCa-based captioner model. The method, called LIST, mixes synthetic and augmented pairs of image and text. Evaluation of the captioner is done against other contrastive models, like CLIP and CoCa.

**Strengths:**

- Formalization of the framework in 3.1 and CoCa in 3.2

**Weaknesses:**

- The overall presentation is hard to follow. Figure 1 shows a captioner model that is "instantiated by Transformers" (L079). It is unclear what that means. Is that a vision-language transformer-based, like LLaVA or Intern-VL2, that is trained on image-text pair captions? If yes, why not stating it clearly? The "Data Engine" is "tasked with generating (...) captions (...) using the captioner model" (L085-L086). What does that mean? There is a captioner model in the box 'Data Engine' in Figure 1, or does the captioner model provide back captions? To make the presentation a bit more confusing, L081 states "a small-scaled paired data augmented by the data engine with the help of LLMs, e.g. LLaMa". Does that mean the data engine is a LLM? Why not making that clear in figure 1? The explanation L079 to L088 is disconnected with that is being shown in Figure 1. Why not making the figure 1 clearly showing what is inside that data engine?
- If the data engine is a LLM, this could weaken importantly the scope of this work. It is unclear how we could call this work self-training when an external component such as LLM is needed to make functioning the data engine.
- Likewise a captioner is used, but it seems the authors (a) use a fairly older captioner design based on CoCa, and (b) do not report any captioning metrics. The evaluation is lacking.
- Nitpick: Use 'multimodal foundation models' (L011) or 'multi-modal foundation models' (L032), but not both. Keep your terminology consistent.
- Nitpick: Use 'Data engine' (L085) or 'Data Engine' (Figure 1), not both.
- L280: The reference is wrong for LLaMa 2. Touvron et al., 2023 is LLaMA.
- While the formalization in 3.1 is a great addition, I wonder if this is necessary for this work and should be instead replaced by a clear picture of the process which is missing at the moment.
- Arguably a good Figure 2 should have been comparison of non cherry-pick captions between LIST, AltText, VeCap, and other concurrent works. The contribution of this work is mentioned to be generating "a diverse range of captions for both paired and unpaired images" (L106-L107). Can we see them? I would suggest the authors to take inspiration from Figure 1 of VeCLIP paper. Or the figures 1 and 3 of [1].
- Nitpick: The formalization of CoCa in 3.2 is not harmful but sounds scholar. Why not referring the reader to the CoCa paper and keeping the space of the paper to show actual examples of captions? The paper is lacking in qualitative comparison with other works, especially captioners. The choice of extending on CoCa architecture is even more surprising that L212 states "LIST is a generic framework and is agnostic to the specific architecture of the vision-language model M".

[1] Lai, Zhengfeng, et al. "Revisit Large-Scale Image-Caption Data in Pre-training Multimodal Foundation Models." arXiv preprint arXiv:2410.02740 (2024).

**Questions:**

- Why using a language encoder along a language decoder for the captioner? Modern captioners commonly use a decoder (>2023).  As well, why not evaluating said captioner? Only ImageNet and VTAB are reported. Evaluation of the captions produced seems an important missing element (e.g. CoCo Captions, TextCaps, etc.)
- I am not exactly clear why this work is stated to be 'Self-Training' when (a) off-the-shelf LLM are needed and (b) the captioner being self-trained is not evaluated. Could you make that clear in the abstract? Commonly Self-Training reuses the model being trained and this is done across multiple iterations (commonly 3), which are each evaluated independently.
- Why not using your captions to train a Multimodal LLM? This would give a clear signal about the quality of your data engine. Something similar to Table 3 of [1] seems importantly needed in this paper.
- Can we see the data? The paper does not show any examples of the data produced. Do you plan to release the dataset?

---

### Official Review · Reviewer_E57m · 2024-11-04

**Soundness:** 3
**Presentation:** 3
**Contribution:** 2
**Rating:** 5
**Confidence:** 4

**Summary:**

This manuscript introduces Language-Image Self-Training (LIST), a semi-supervised framework designed to enhance the alignment of multi-modal (e.g., vision and language) models by leveraging unimodal data (without text modality). For a given unimodal data (image or audio), the proposed method leverages the captioner model with LLM-based refinements and performs multi-modal learning on generated paired data. Throughout extensive experiments, LIST demonstrates its effectiveness on various benchmarks, including zero-shot classification, retrieval, and even audio-language alignments.

**Strengths:**

- This manuscript addresses the important problem of training from unpaired unimodal data. But I think the word ``unpaired'' is a little bit confusing as I first imagine leveraging unpaired image and text data, not unimodal image data.

- This manuscript reports notable performance gains on various vision-language benchmarks including zero-shot classification, image-text retrieval, and so on. The consistent improvements across multiple benchmarks validate the effectiveness of LIST.

- LIST demonstrates strong generalization capabilities not only for vision-language tasks but also for audio-language representation, which suggests the potential applicability of this approach across other modalities.

**Weaknesses:**

- I feel this approach heavily relies on the large language models. Without LLM refinement, is it possible to generate reasonable captions when small paired data is given? I think the contribution of LLM-based caption refinement is not well-validated.

- There is no heterogeneousness analysis between paired data and unimodal data. It is straightforward that training would be struggled if the unimodal images are not closely seen via paired data.

- There is no direct comparison between LIST and a simpler approach using a pre-trained captioner (including MLLMs) without the iterative data refinement process. This makes it difficult to demonstrate the actual contribution of the LIST-specific refinement loop. Moreover, the LIST framework's reliance on repeated caption generation, especially involving LLMs, implies huge computational costs. The authors lack an analysis of the trade-offs involved and do not compare LIST's computational efficiency to naive alternatives; for example, one can generate captions for unimodal data using pre-trained captioners and do CoCa training.

- Although the authors claim significant improvements, I think comparisons with CoCa trained on a small number of data (160 or 600K) are quite unfair. Nevertheless, I found consistent improvement compared to CoCa trained on full paired data (2.8M). Interestingly, the proposed method even outperforms the full CoCa on SugarCrepre. I guess it is due to the high-quality captions generated by LLMs. It would be great if the authors could validate LIST's ability without LLM refinements, which are computationally expensive.

- The impact of the backbone Model (DINOv2) is unclear. Although the authors mentioned the use of DINOv2 in section 3, there are no mentions in Tables about whether it uses DINOv2 or the impact of DINOv2. When considering CoCa is trained from scratch and DINOv2 is the state-of-the-art SSL model, it would be necessary to show the impact of DINOv2 in the LIST improvements for a fair comparison.

**Questions:**

Please see the weakness section.

---

### Note · Authors · 2024-11-13

I have read and agree with the venue's withdrawal policy on behalf of myself and my co-authors.